# New Design and Optimization of a Jet Pump to Boost Heavy Oil Production

**Jens Toteff [1], Miguel Asuaje [1,2,*] and Ricardo Noguera [3]**

1   Energy Conversion Deparment, Simón Bolívar University, Caracas 89000, Venezuela; jtoteff@gmail.com
2   Frontera Energy, Operations New Technology Department, Calle 110 # 9–25, Bogota 110111, Colombia
3   École Nationale Supérieure d'Arts et Métiers, LIFSE—Arts et Métiers, 75103 Paris, France;
    ricardo.noguera@ensam.eu
*   Correspondence: asuajem@usb.ve

**Abstract:** In the Oil and Gas industry, installing pipe loops is a well-known hydraulic practice to increase oil pipeline capacities. Nevertheless, pipe loops could promote an unfavorable phenomenon known as fouling. That means that in a heavy oil-water mixture gathering system with low flow velocities, an oil-water stratified flow pattern will appear. In consequence, due to high viscosity, the oil stick on the pipe, causing a reduction of the effective diameter, reducing handled fluids production, and increasing energy consumption. As jet pumps increase total handled flow, increase the fluid velocities, and promote the homogenous mixture of oil and water, this type of pump could result attractive compared to other multiphase pump systems in reactivating heavy crude oil transport lines. Jet pumps are highly reliable, robust equipment with modest maintenance, ideal for many applications, mainly in the oil and gas industry. Nevertheless, their design method and performance analysis are rarely known in the literature and keep a high experimental component similar to most pumping equipment. This paper proposes a numerical study and the optimization of a booster multiphase jet pump system installed in a heavy oil conventional loop of a gathering system. First, the optimization of a traditionally designed jet pump, combining CFD simulation and optimization algorithms using commercials software (ANSYS CFX® and PIPEIT® tool), has been carried out. This method allowed evaluating the effect of multiple geometrical and operational variables that influence the global performance of the pump to run more than 400 geometries automatically in a reduced time frame. The optimized pump offers a substantial improvement over the original concerning total flow capacity (+17%), energy, and flow distribution. Then, the effect of the three jet pump plugin configurations in a heavy oil conventional trunkline loop was analyzed. Simulations were carried out for different driving fluid pressures and compared against a traditional pipeline loop's performance. Optimum plugin connection increases fluid production by 30%. Finally, a new eccentric jet pump geometry has been proposed to improve exit velocities and pressure fields. This eccentric jet pump with the best connection was analyzed over the same conditions as the concentric optimized one. An improvement of 2% on handled fluid was achieved consistently with the observed uniform velocity field at the exit of the pump. A better total fluid distribution between the main and the loop line is obtained, handling around half of the complete fluid each.

**Keywords:** CFD; jet pump; heavy oil field; oil production

## 1. Introduction

Mature heavy oil fields are frequently characterized by a total fluid production with high water cut (under 90%). Excessive water production causes incremental costs, energy consumption, and inefficient oil recovery from mature oil fields. Hilly terrain topography, long distances in trunk lines generate significant hydraulic imbalances, creating restrictions to the flow of fluids to the processing center. Due to high rates of surface water, one of the associated problems is that, frequently, pipelines maximum capacity is reached at the early

production times in the field. To improve their performance and increase the flow capacity of these lines, a very well-known and familiar solution is installing pipeline loops, as shown in Figure 1. The purpose of a pipe loop installed in a segment of a pipeline is to reduce the amount of pressure drop in that section of pipe by increasing the equivalent diameter.

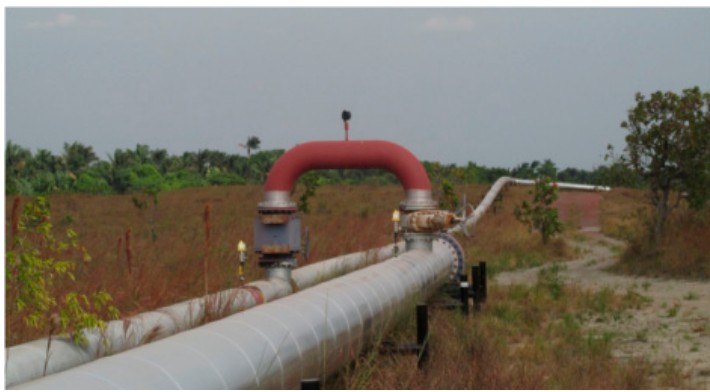

**Figure 1.** Loop in pipelines of heavy crude oil.

However, when handling multiphase heavy crude oil and water flow, a common problem appears after the loop is built. Due to low fluid velocities, a phases separation would cause an effect contrary to the desired when constructing the loop. This effect is known as fouling, and several authors have shown how damaging it is to the optimal transport of hydrocarbons [1]. Figure 2 shows the effect of the pressure increase in a pipeline with heavy crude oil transport. In transporting heavy crude oil with high water content (Wcut > 90%), such as mature fields, pressure losses increase monotonically as the pipeline fouls. Then, a high blockage is experienced, causing very high energy consumption and a substantial restriction for wells fluids apports due to back pressure on trunk lines.

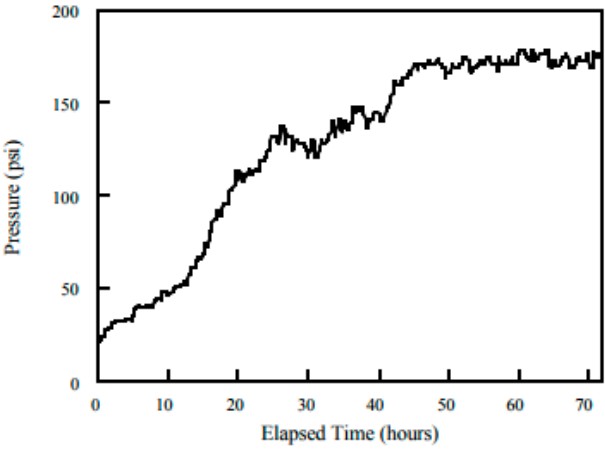

**Figure 2.** Fouling of San Tomé test loop with Zuata crude oil Venezuela [1].

Figure 3 shows the fouling phenomenon, highlighting low fluid velocities (Figure 3a) and phase separation downstream of the loop connection point through CFD [2]. In the loop line, velocities below 0.5 m/s have reached, promoting the loop's work as a gravity separator (Figure 3b).

Due to its characteristics, and as sequence of previous works [2,3], this paper proposes using a jet pump as a multiphase booster pump for mature heavy oil fields. With this type of pump to goals can be reached:

1.  Significant upstream pressure reduction can be obtained taking advantage of the pump operational principle (Venturi effect).
2.  Downstream velocity increases at the pipeline employing the driven fluid.

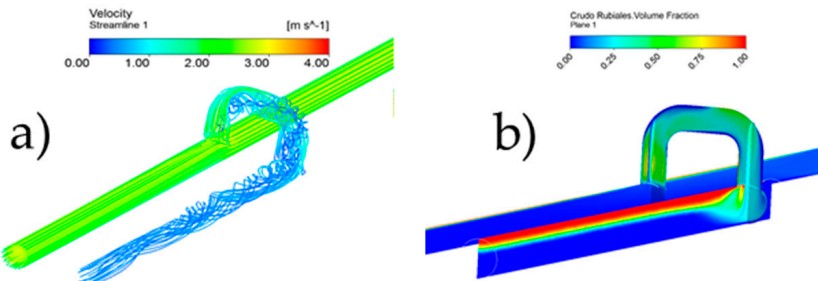

**Figure 3.** Multiphase CFD simulation in a pipeline loop, (**a**) Velocities streamlines, (**b**) Oil fraction [2].

Starting with a pump designed by a traditional method, an optimized geometric was obtained, evaluating the effect of multiple geometrical and operational variables by a combined methodology using CFD simulations and optimization algorithms. Then, the impact of three plugin configurations of the jet pump in a conventional trunkline loop was analyzed. Finally, a new eccentric jet pump geometry has been proposed to improve exit velocities and pressure fields.

## 2. Jet Pump Theory

Ejectors (also known as Jet pumps) are devices with no moving parts using fluids under controlled conditions. With a high-pressure driving flow, they boost a low-pressure flow discharging at intermediate pressures. The operation principle converts the motive fluid's total pressure (primary fluid) into velocity through a nozzle. The high speed creates a low-pressure zone in the suction chamber causing secondary fluid to be pumped into the suction chamber. The two liquids are mixed by exchanging momentum in the ejector's neck. Total mixture flow goes into the diffuser, where hydraulic energy is recovered until an intermediate pressure.

Figure 4 shows the parts and principal dimensions of the ejector. These are:

(a) Nozzle: entrance of the high-pressure driving fluid. It will convert pressure into velocity, generating a low-pressure zone and the secondary fluid's suction (in the annular).

(b) Suction chamber: is the entrance of pumped fluid and where the nozzle is arranged.

(c) Throat: is the section, usually of constant diameter, where the mixing and kinetic energy transfer of the driving and sucked fluids is carried out, forming a uniform velocity profile.

(d) Diffuser: is the discharge section of the pump. It is in charge of energy recovery, reducing the velocity, and increasing the static pressure of an incompressible fluid passing through it.

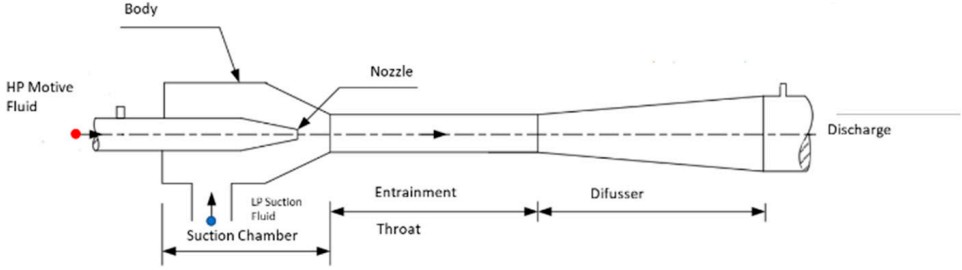

**Figure 4.** Jet pump diagram [2].

Gosline suggested jet pump theory in 1934 [4], establishing the governing equations. It is developed from two essential fluid mechanics equations: Bernoulli and linear mo-

mentum equations [5]. Along a streamline and, after some simple manipulations, two nondimensional parameters are defined:

$$N = \frac{H_{disc} - H_{LP}}{H_{HP} - H_{disc}}, \tag{1}$$

$$M = \frac{Q_{LP}}{Q_{HP}} \tag{2}$$

$N$ is the hydraulic energy ratio, and $M$ is the flow ratio.

Pump efficiency $\eta$ is defined as the ratio of total energy increase of suction flow to the total energy transfer by the driving flow:

$$\eta = M \cdot N \tag{3}$$

After introducing the linear momentum equation, design methodology from Karassik's [5] propose the following equation to predict the performance of a Jet pump:

$$M = \sqrt{N} - 1 \tag{4}$$

A correction factor $\in$ will consider energy losses associated with mixing both streams, friction, and diffusion losses. $\in$ is estimated by empirical curves. Using the above equation, either motive flow rate or high pressure could be calculated for suction operating parameters.

$$M = \in \cdot \sqrt{N} - 1 \tag{5}$$

Nozzle and diffuser section area and diameters are estimated using the continuity equation.

$$d = \sqrt{\frac{4Q}{\pi V}} \tag{6}$$

Many researchers such as Cunningham and River [6], Cunningham [7], Teaima and Meakhail [8], Aoiki, Prabkeao [9], Hammoud [10], Saker and Hassan [11] worked to maximize jet pump's efficiency, adding corrective terms and reporting experimental optimum values for different geometrical parameters such as $l_n/d_t$, $l_n/l_t$, $l_n/d_n$, among others. They try to find the optimal geometrical configuration, varying parameters such as those shown in Figure 5 [2].

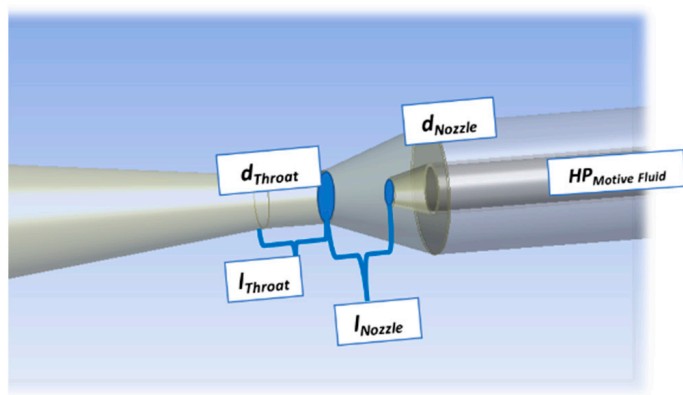

**Figure 5.** Variables involved in the optimization process [2].

## 3. Methodology

### 3.1. Booster Pump in a Loop

For an oil production looped gathering system, a custom-designed ejector can operate as a booster pump assisted by a smaller single-phase pump providing a high-pressure

motive fluid. Figure 6 shows a simplified scheme of the pipeline loop with and without a booster jet pump.

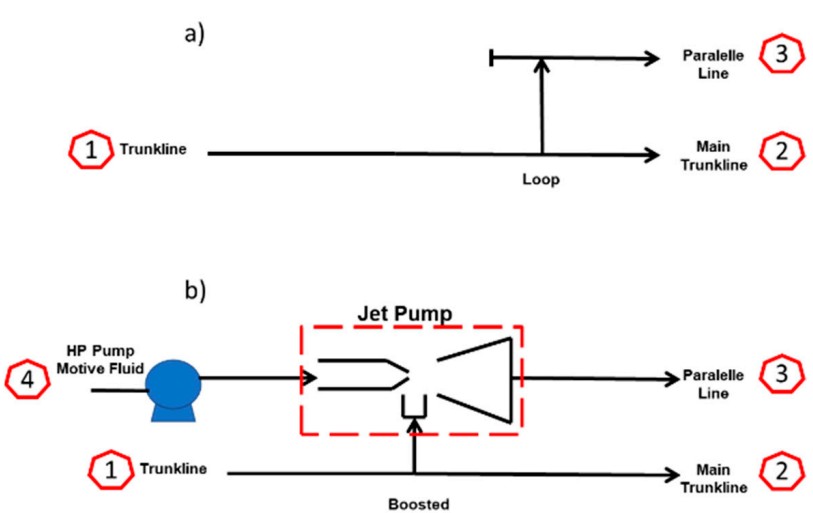

**Figure 6.** Booster jet pump installed in a trunkline loop scheme. (**a**) without jet pump, (**b**) with jet pump [3].

First, a jet pump has been designed through the conventional method [5]. Once the main dimensions have been fixed to this first pump, called seed pump, a first CFD simulation was conducted to study its performance over the simulation domain shown in Figure 7.

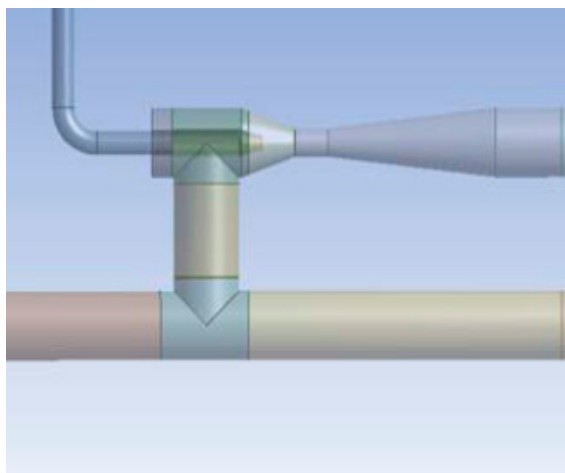

**Figure 7.** Simulation domain of jet pump.

### 3.2. Jet Pump Design and Optimization

An optimization routine to iterate over the main dimensions, combining their performance analysis through the CFD, was implemented to design an optimal jet pump configuration. CFD commercial code ANSYS CFX® (v-14, Canonsburg, PA, USA) and the optimization algorithms of the Pipeit® (Version 1.0, Throdheim, Norway) tools have been used. Pipeit® is an Integrate and optimization software that originally conceded to integrate models and optimize petroleum assets. As an integrator, Pipeit® allows to chain together an unlimited number of applications, run sequentially and in parallel [12]. The proposed methodology is schematized according to Figure 8. After fixing the variables, constraints, and the objective function, using the IBM non-linear IPOPT (Interior Point Optimizer for large-scale nonlinear optimization) optimization solver, several geometrical

combinations for the pump were obtained, and its performances were predicted through CFD simulations [2].

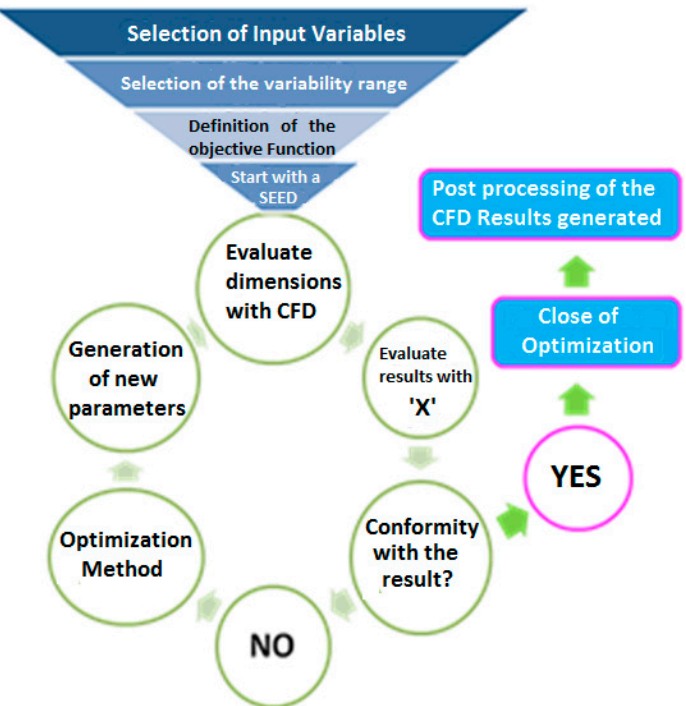

**Figure 8.** Jet pump dimensioning and optimization methodology [2].

Keeping in mind that the objective of the jet pump of the present case study is to increase the total flow capacity of fluid handled by the main and the loop pipeline system, a dimensionless flow function was defined as follow:

$$X = \left| \frac{|50 - Q_3^*|}{50} - 1 \right| \cdot \frac{Q_1}{Q_4} \tag{7}$$

where:

$$Q_3^* = 100 \cdot \frac{Q_3}{Q_1} \tag{8}$$

The new variable $X$ proposes to be an objective function that will maximize the total flow in the system, minimizing the jet pump driving fluid $Q_1$ guarantying the evenest flow distribution. $X$ value will be maximum when the derived flow $Q_3^*$ tends to be 50% of the total flow, ensuring an even flow distribution in the gathering system. The flow ratio between the primary and driving fluid lines will be maximum as lower high-pressure motive fluid is used.

### 3.3. Analysis of Plugin Configurations

After obtaining the optimal pump internal geometry, a study concerning the influence of plugin connection in the loop was carried out. Those studies aimed to improve intake flow conditions. Figure 9 shows the original pipeline loop without a jet pump and three different configurations proposed to install the ejector. The first configuration is the standard 90° elbow from the mainline, which is the most straightforward construction. Options 1 and 2 represent an improvement by deriving the flow from the main lines with a special 30° fitting [3], which promotes better hydraulic behavior. Nevertheless, its construction and installation will represent a challenge over the standard configuration.

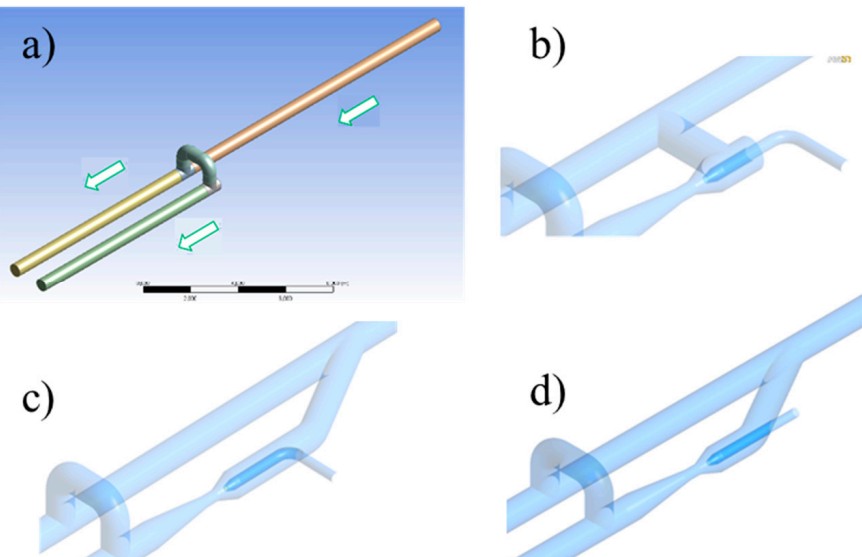

**Figure 9.** Fluid physical domain (**a**) Original loop, (**b**) Jet pump standard installation, (**c**) Jet pump option 1, (**d**) Jet pump option 2 [3].

*3.4. New Proposed Geometry. Eccentric Jet Pump*

Despite the improvement over the geometry and suction conditions, a non-uniform flow field was detected at the pump discharge after these first stages of the study. Consequently, a geometric adjustment that involves designing the ejector housing eccentrically (Figure 10) was proposed to improve this condition. This new eccentric jet pump maintains the exact internal geometric dimensions of the concentric optimized one.

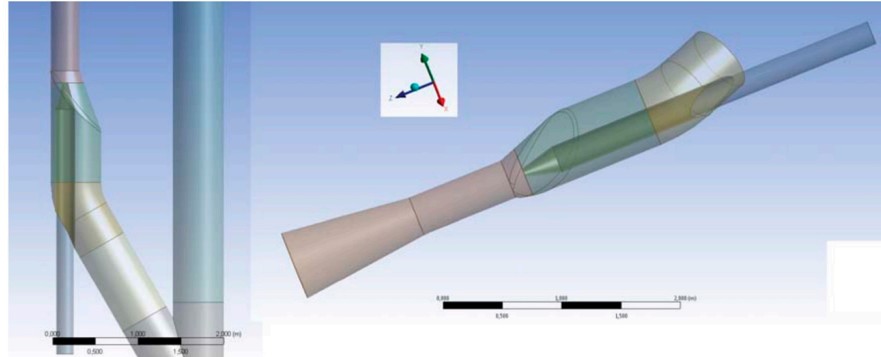

**Figure 10.** New eccentric jet pump.

## 4. CFD Modelling

A rigorous CFD study was carried out over all pump geometries to estimate its performance and analysis its internal flow fields, from the seed pump to the optimized one. All simulations were carried out through commercial software ANSYS®CFX. A template to control ANSYS from geometry generation, meshing, solving, and post-processing was built in Pipe it®.

*4.1. Grid*

The fluid domains of the loop and the jet pump fluid used for simulations have been shown in Figures 7, 9 and 10. Pipelines nominal diameter are 24 inches. The total pipe's length was 25 m.

Non-structure grid with inflation layers were sat up for all meshes. An independent mesh study was carried out. Three mesh sizes were used: The coarser mesh was made

with 2,100,000, the medium grid with 3,800,000, and the fine grid with 6,400,000 number of elements. Simulations were performed over different inlet boundary conditions, obtaining associate errors of less than 1%. More details are reported in Toteff, 2015 [13]. Figure 11 shows an example of the grid.

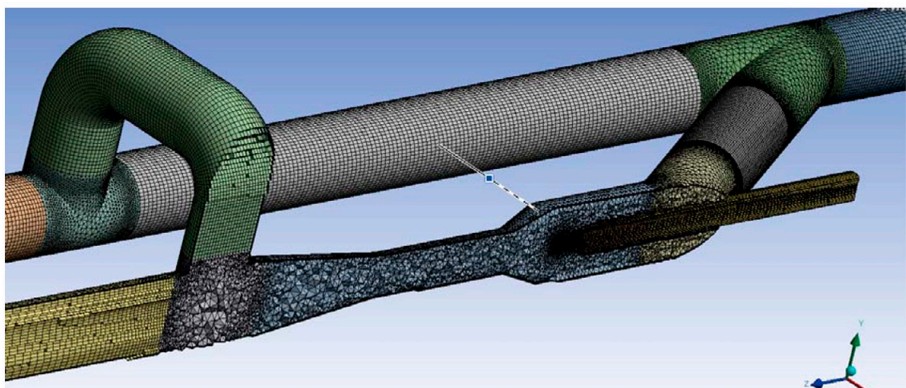

$l_n$ = 481mm; $d_t$ = 336mm; $l_t$ = 1235mm; $d_n$ = 40mm

**Figure 11.** Mesh of the Jet pump.

### 4.2. Fluids Properties and Numerical Approach

For the design and optimization stage, all performance analysis was performed by steady-state water single-phase flow simulations. Water was considered as an incompressible and isotherm flow under gravity effects.

For the plugin analysis, all simulations were carried modeling the system using a homogeneous oil-water mixture as fluid, with a water cut of 94%. The jet pump motive fluid is water. Euler-Euler approach was used considering oil droplets dissolved in water were considered as a homogeneous mixture. As suggested by Aldas and Yapici [14], SST turbulence model is used for the continuous phase due to its ability to predict boundary layer behavior. The SST turbulence model is a combination of the κ-ε model in the region outside the boundary layer and κ-ω in the inner zone of the boundary layer. Water and oil properties are shown in Table 1. A summary of numerical conditions is shown in Table 2.

**Table 1.** Fluid Properties @ 25 °C.

| Property/Fluid | Water | Oil |
|:---:|:---:|:---:|
| $\rho$<br>Density (kg/m$^3$) | 997.00 | 974.78 |
| $\mu$<br>Viscosity (cp) | 1.00 | 277.45 |

**Table 2.** Summary of simulation conditions.

| Parameter | Details |
|:---:|:---:|
| Fluids | Water/Water–Oil |
| Wcut | 94% |
| Turbulence model | SST |
| Solving | Steady State |
| Advection Scheme | 2nd Order |
| Stop criteria | For mass and momentum: RMS < 1 × 10$^{-4}$<br>and unbalances < 2%. |

The boundary conditions are presented in Table 3. Low total pressure at main pipeline suction and outlet pressures were kept constant. The high-pressure motive fluid which represents, the energy supplied to the ejector, was varied.

**Table 3.** Boundary conditions.

| Number | Parameter | Condition |
|:---:|:---:|:---:|
| 1 | Main Pipeline Suction Fluid Inlet LP | Total Pressure (100 psia) |
| 2 | Main Pipeline Outlet Flow | Static Pressure (98 psia) |
| 3 | Loop Outlet Flow | Static Pressure (98 psia) |
| 4 | Motive Fluid Inlet HP | Total Pressure (Variable) |

## 5. Results

### 5.1. Original Jet Pump and Optimized

The proposed methodology allowed evaluating many geometric combinations of Jet pumps in a reasonable time frame of fewer than two weeks. This study was solved on an I7 computer with 64 GB of RAM and 2 TB of memory. This performance analysis study would be impossible to do experimentally. Dimensions and operational conditions for the seed pump and optimized one (Pump 1) are shown in Table 4. As presented, the optimized geometric is entirely different from the original one. The position of the nozzle has been considerably reduced, and the throat diameter increased. Mixing zone length or throat length has been double. Only the nozzle diameter had a minor change, around 20% of the original dimension. This new geometry is consistent with a reduction in driven flow and a better mixing zone. That will result in an 8% increment of total handled fluid with a significant power reduction, guaranteeing an even flow distribution at the loop exit.

**Table 4.** Dimensions of the pumps.

| Variable | Seed Pump | Optimized Best Pump 1 |
|:---:|:---:|:---:|
| $l_n$ (mm) | 1000 | 354 |
| $l_t$ (mm) | 500 | 1114 |
| $d_n$ (mm) | 50 | 40 |
| $d_t$ (mm) | 152 | 337 |
| $P^T_4$ (psi) | 135 | 110 |
| X (-) | 8.14 | 49.95 |
| $Q_1$ (kBFPD) | 428.9 | 464.7 |
| $Q_4$ (kBFPD) | 96.0 | 7.4 |
| $Q_{2+3}$ (kBFPD) | 525.0 | 472.1 |

Figure 12 shows the result from almost 400 geometric combinations proposed by the optimization program and simulated by CFD software. This figure illustrated the evolution of the objective function against geometrical and operational variables. Red points represent the pumps configurations with results in and exit flow distribution almost symmetrical, which means, *X* value is bigger than 45. The green and yellow points correspond to the optimum pump geometries, implying the objective function has at least two local maximums. Again, as can we appreciate in this figure, dimensions between best pump 1 and 2 have no significant differences.

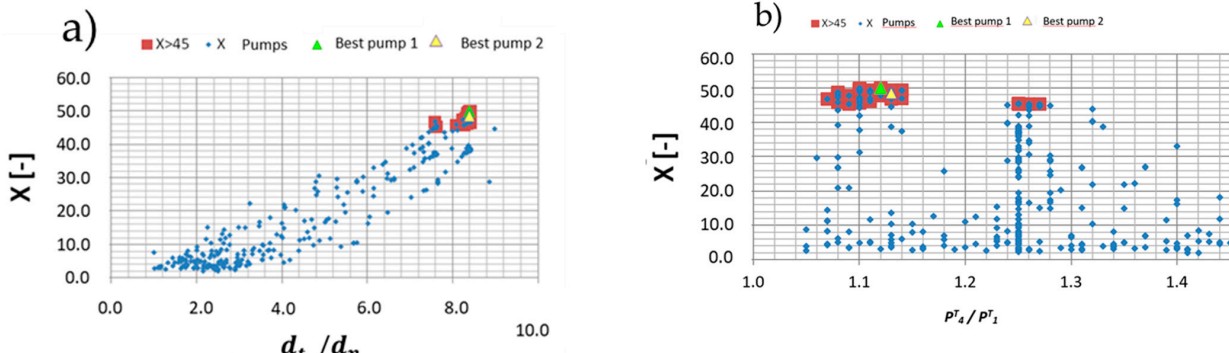

**Figure 12.** (**a**,**b**) Objectif function behavior against geometric and operational parameters.

These two pumps (Best pump 1 and 2) have similar geometries, but they are pretty different from the other analyzed pumps [13]. These configurations will increase flow velocities and considerably reduce the oil fouling effect on the pipe. As a function of operational and geometrical parameters, the maximum of the objective function has a clear trend for some variables. As observed (Figure 12a), there is a clear positive linear tendency to increase $X$ as the $d_t/d_n$ ratio. On the other side, the direction $X$ as a function of operational and other variables is not conclusive. For example, a clear trend in achieving a maximum value is not observed in the curve $X$ vs. pressure ratio (Figure 12b). That means the optimal $X$ value, which implies an even flow distribution, will directly relate to the operating pressure ratio between the driving and the boosted fluids. Consequently, a compromise between a minimum pressure ratio and an even flow was made to select the optimal pump. As expected, a lower pressure ratio results in lower power consumption to boost and drive the fluid. More details and the exact dimensions of best pump 1 and 2 can be found in Toteff 2015 [13].

Performance analysis for different driving fluid conditions was carried out for the selected geometric configurations Best pump 1 and Best pump 2. The main results are shown in Figures 13–15.

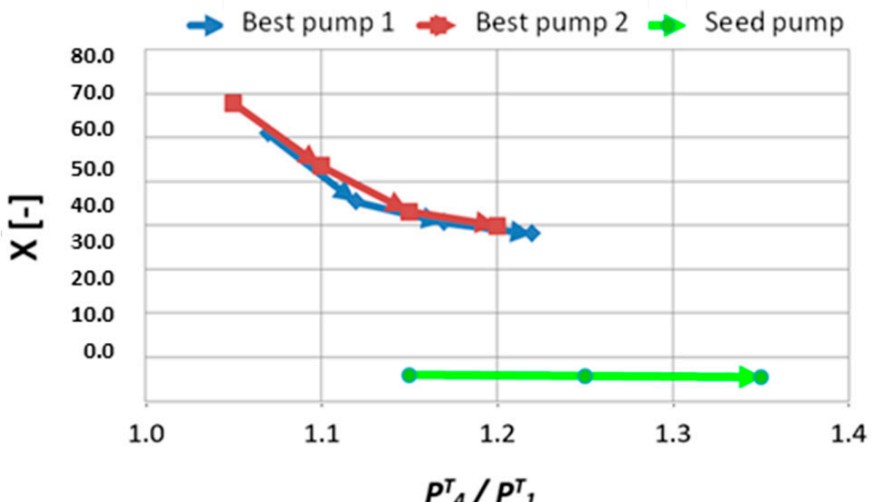

**Figure 13.** $X$ vs. $P_4^T/P_1^T$.

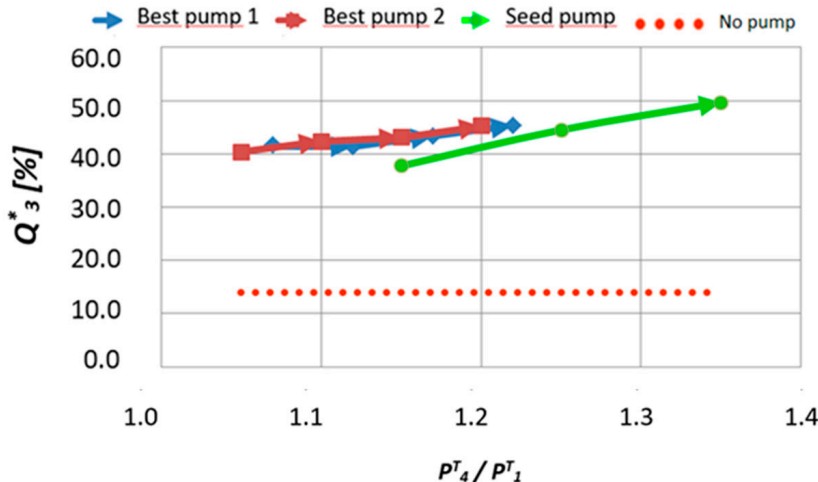

**Figure 14.** $Q_3^*$ vs. $P_4^T/P_1^T$.

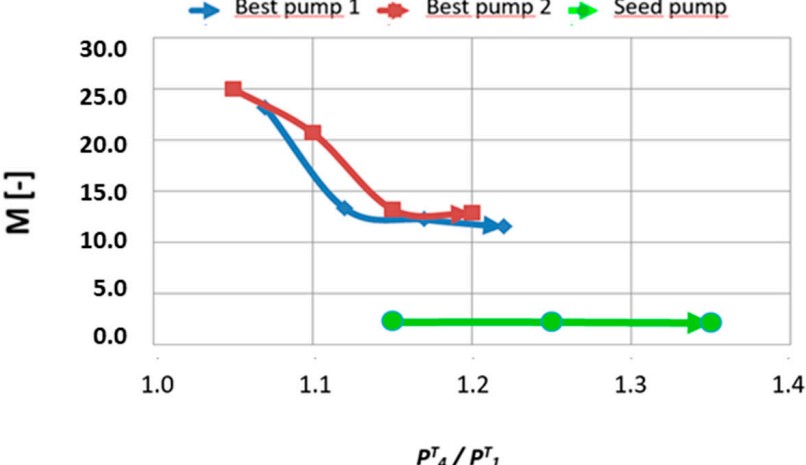

**Figure 15.** $M$ vs. $P_4^T/P_1^T$.

As expected, both pumps have similar behavior. The new geometries achieve a more significant value of $X$ compared to the seed pump. $X$ will be between 40% and 70%, while the conventional pump reports much lower values, around 8. That means that total handled fluid will be increased using the new pumps. All the pumps obtain an even distribution of flow in the lines, as is being shown plotting nondimensional outlet flowrate $Q_3^*$ vs. pressure ratio. That validated that jet pump has a good performance avoiding unfavorable flow distribution which promotes the pipelines fouling phenomena on jeopardy of oil production.

In addition, optimized pumps require lower pressure ratios to reach this even flow distribution, which means low energy consumption. By way of reference, the figure shows the flow that the system would handle without the pump, which is one-third of the value using a booster pump.

Figure 15 shows $M$ variable, for the new pumps and the seed one. In this case, it does not make sense to estimate the pump's efficiency similar to previously defined $\eta = M \cdot N$, because for all the simulations $N$ value is constant and has been fixed as a boundary condition. As shown in Figure 15, M decrease as $P_4^T/P_1^T$ increase as consequence of the increase of total fluid. Increasing pumping fluid or reducing driving fluid will promote the separation of the boundary layer in the diffuser that generates significant recirculation and energy losses. This fact is evidenced in the following images (Figure 16), showing velocity fields inside the original pump and the best pump 1. Velocity fields delight the

better performance for the optimized one. Figure 16a shows a higher zone of driving fluid for the seed pump, which jeopardizes its performance and the total fluid derived from the mainline. Contrarily, the optimal pump (Figure 16b) in the driving fluid section is smaller, showing the driving fluid's optimization and, consequently, power.

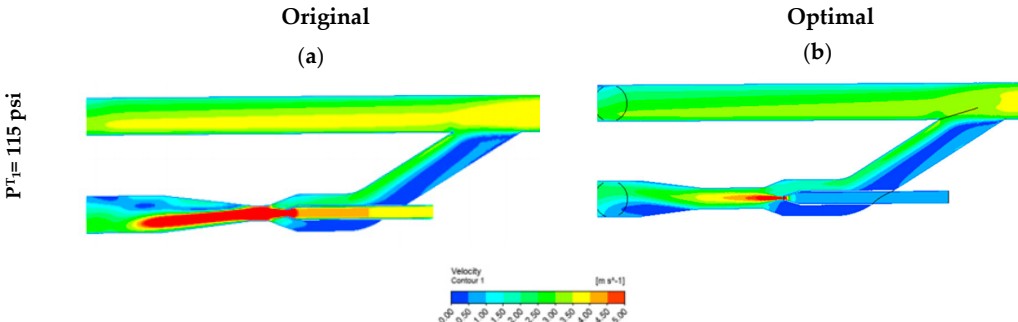

**Figure 16.** Velocity contours for original seed pump (**a**) and optimized one (**b**) @ 115 psi.

*5.2. Plugin Connection Analysis*

The effect of three different plugin configurations (Figure 9) on the pump's performance was studied. A nondimensional inlet flowrate was defined to quantify the impact of installing the booster pumps, as the ratio between the inlet flow rate with booster pump divided by the flow rate handled only by the loop without any pump. This parameter is:

$$Q_1^* = \frac{Q_{1\ Booster}}{Q_{1\ Loop}} \tag{9}$$

Figure 17 shows the percentage of increase of total flow rate by using a booster jet pump. A standard 90° connection offers a narrow performance compared against de plugin options 1 and 2. The range of incremental fluid would be between 5 to 9% overall pressure ratio range. Straitened behavior on standard jet pump connection can be explained by the diffuser's non-uniform velocity field, as shown in Figure 18. This exit recirculation results from a non-uniform velocity inlet into the suction chamber. Note that regardless of pressure, there is a strong recirculation at the low-pressure ejector intake. So, the suction area's effectiveness is significantly reduced, causing a flow restriction for all high-pressure conditions. In this configuration, the low-pressure fluid flow has radial entry into the throat, which produces a deflection in the motive fluid line, causing prematurely fluid flow detachment in the diffuser.

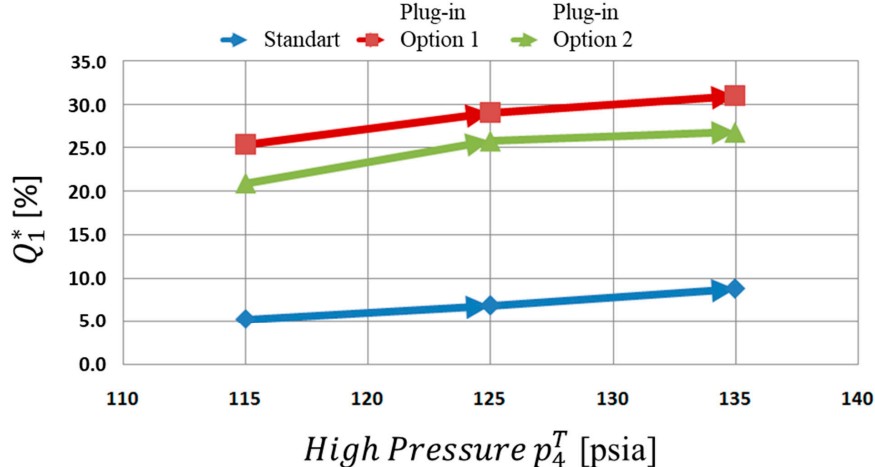

**Figure 17.** Nondimensional Flow Rate in the loop using a Jet Pump.

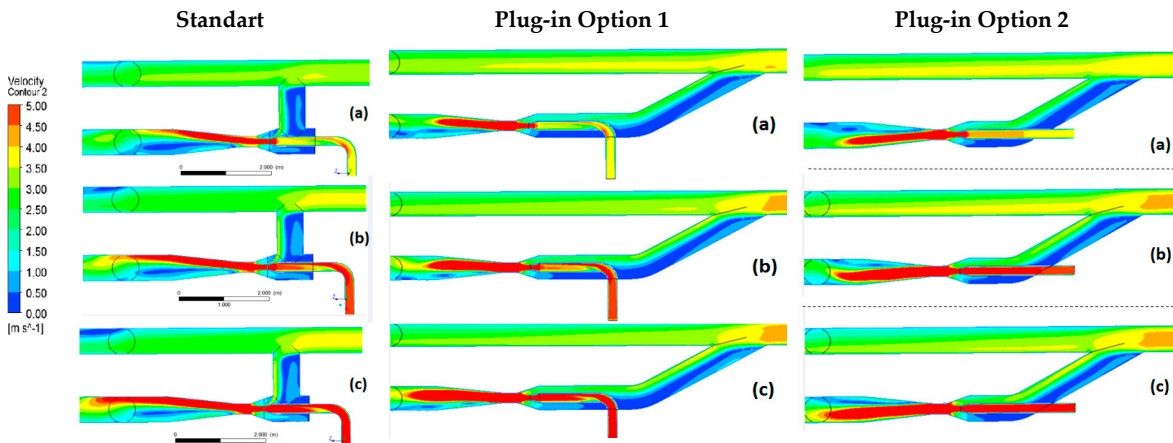

**Figure 18.** Velocity fields for Jet Pump standard installation at different HP inlet conditions (**a**) 115 psia, (**b**) 125 psia, (**c**) 135 psia.

On the other hand, for options 1 and 2, using a 30° fitting provides better performance increasing the total flow rate to 30% over the standard connection. Both plugin connections seem to significantly reduce the effect of asymmetry and vortices at the inlet, improving the overall ejector performance. Even if low velocities at the low-pressure side and recirculation at the diffuser are still present, they do not block the total fluid passing through the pump. The incremental fluid range is enlarged, varying up to 7%. In consequence, 30° fitting is recommended for a flow bypass.

*5.3. Eccentric Pump Single Phase Performance Analysis*

Despite the improvement at the intake obtained by the plugin configuration, low-speed internal recirculation continued to exist in many cases in the suction chamber, blocking the suction flow. Looking for an improvement at the inlet velocity field, a new geometric was proposed, consisting of an eccentric suction chamber. The eccentric pump is shown in Figure 19. Internal dimensions and distance were kept equal to the optimized concentric pump to quantify the effect of the suction chamber changes. This new pump was studied by observing the maximal operational for a trunkline condition in the Quifa field where the maximum flow rate is limited between 450 and 500 KBFPD [15]. As well as in previous studies CFD technique is being proposed to optimize the hydraulic performance of equipment in Quifa Field [16]. The hypothesis is that the pump's eccentricity will help reduce the pressure drop at the suction chamber, becoming uniform the velocities at the inlet and the mixture zone.

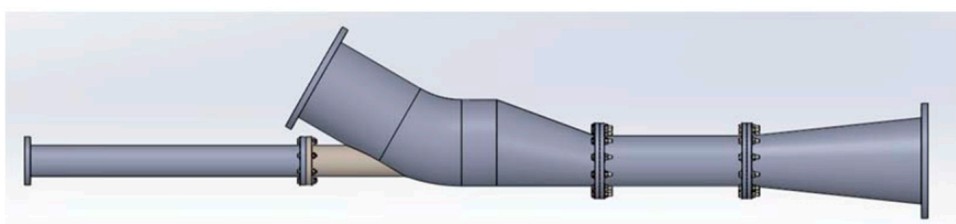

**Figure 19.** Eccentric Pump.

Figure 20 illustrates the flow rates handled by optimized jet pumps concentric and eccentric for different motive fluid operational conditions. As a reference, single-loop values are also shown. In general, while the single loop only ran over 400,000 barrels, an increment of 15% on average is obtained using the booster system. The driving flow rate is less than 3% of the total rate, representing a substantial driving power reduction for all cases.

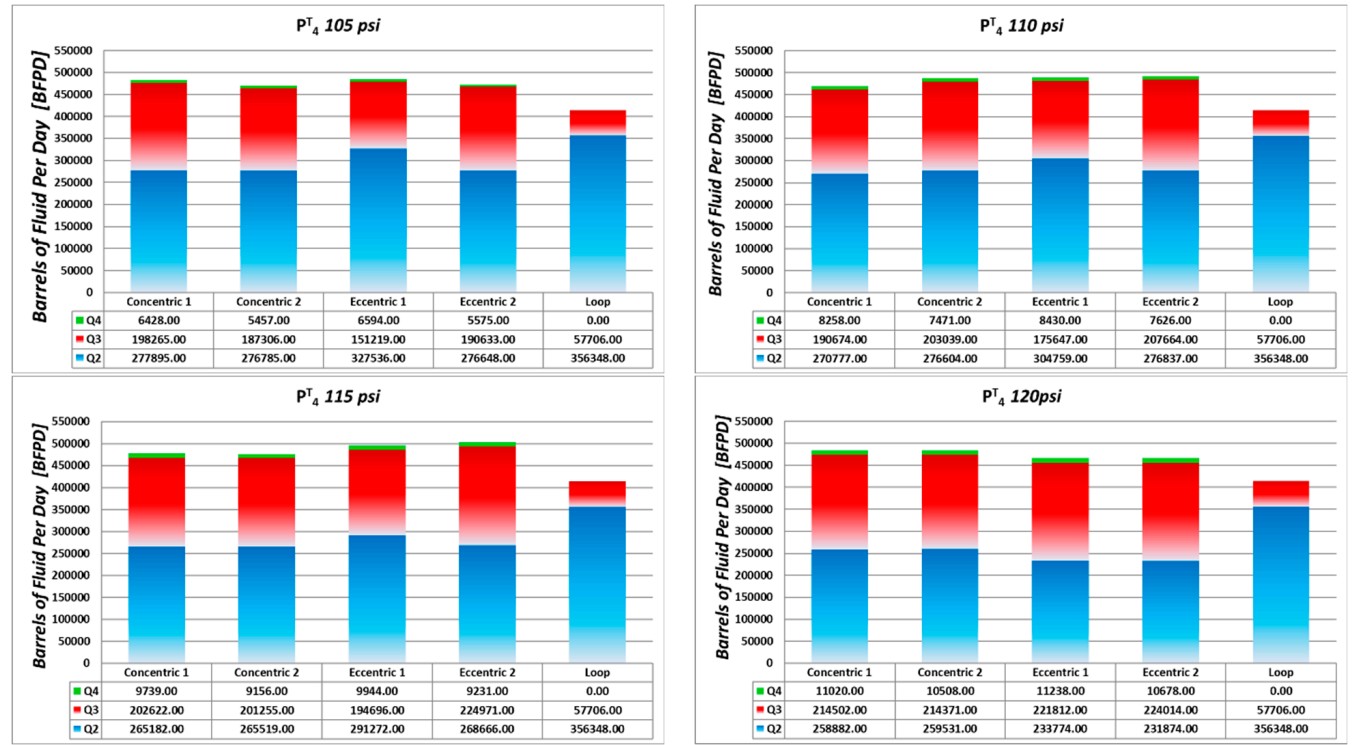

**Figure 20.** Flowrate for concentric and eccentric pumps. Case of study Quifa Field.

Figure 21 shows global performance parameters $X$ and $Q_1^*$ as a function of pressure ratio. The objective of the pump is, of course, to increment total fluid with a minimum driving flow. However, it must also guarantee the evenest possible flow distribution in the lines. These two effects can be seen by analyzing these curves.

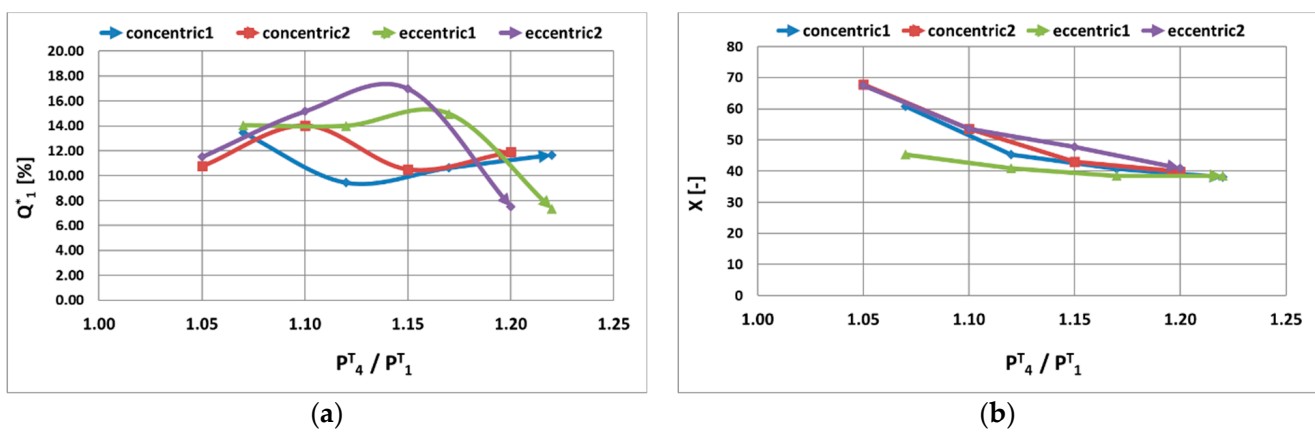

**Figure 21.** (**a**,**b**) Single-phase Eccentric Jet pump vs. Concentric Jet pump Case of study Quifa Field.

Compared against a single loop, Figure 9a, total flow rate increment refereed to a single loop without ejector. Eccentric jet pump 2 shows a maximum incremental of 17% at 115 psi (Figure 21a). In general, this eccentric pump performs better than the other ones, despite the drop of handling flow when the driving pressure increases. Concerning the flow distribution, $X$ values are plotted in Figure 21b. Eccentric ejector 1 guarantees a uniform distribution for all operational ranges, always $X$ greater than 40, and can even reach 45 depending on the working pressure.

Figure 22 shows $M$ behavior which would be equivalent on this case to the efficiency. It can be kept in mind that $N$ values have been imposed on the simulations. The eccen-

tric pump seems to perform better in all the driving pressure ranges. Nevertheless, the difference is only in one simulation point, which should not be taken as a conclusive result. Qualitative results in the velocities field will confirm that the eccentric pump will perform better (Figure 23). As expected, the eccentric jet pump shows a better and uniform flow velocity field which is compared against the concentric one. Inlet blockage has been mitigated, and in consequence, flow exit recirculation was reduced too. That is a point that should be validated by performing experimental work.

$$M = \frac{Q_{LP}}{Q_{HP}} \tag{10}$$

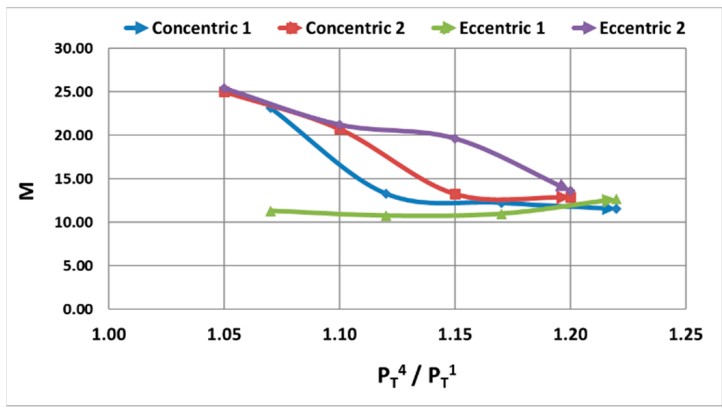

**Figure 22.** Driving fluid on the jet pump vs. pressure ratio.

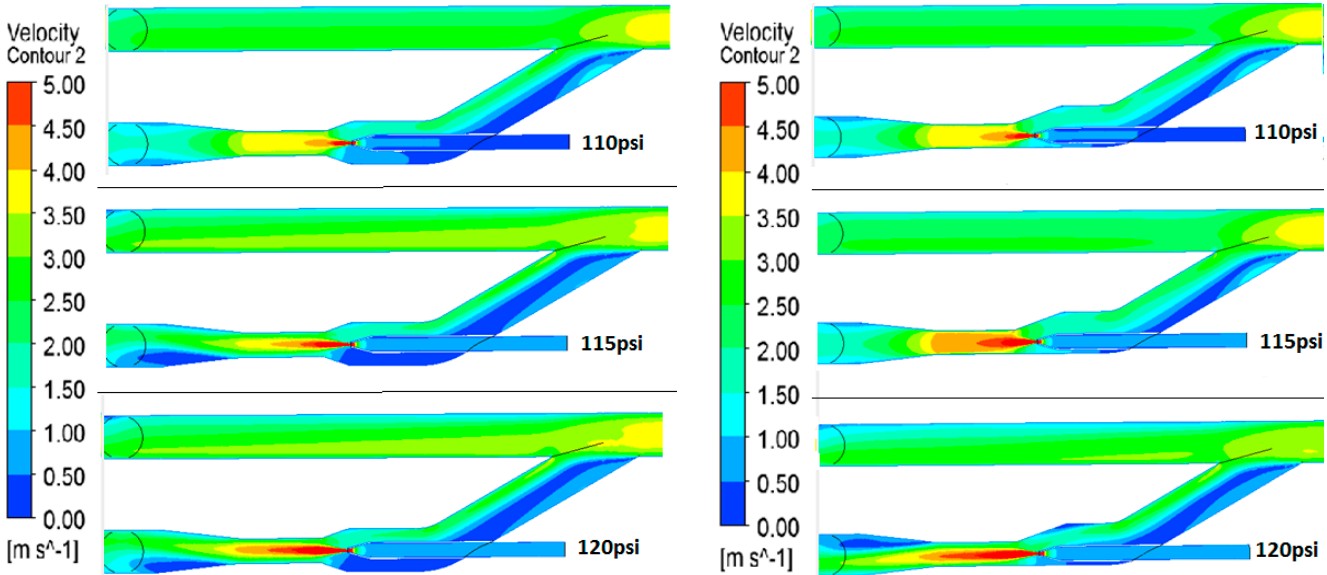

**Figure 23.** Velocity contours for concentric and eccentric pump. Single-phase simulation.

### 5.4. Two-Phase Flow Simulation

Due to the enormous computational cost, the two-phase flow simulation was carried out only after selecting the best pump possible for the case of study.

Figure 24 shows the uneven flow distribution between the main and the loop line. Low velocities and high vorticity made the loop line work as a separator, promoting fouling phenomena in the system. After installing the jet pump, a favorable flow distribution is appreciated on the gathering system.

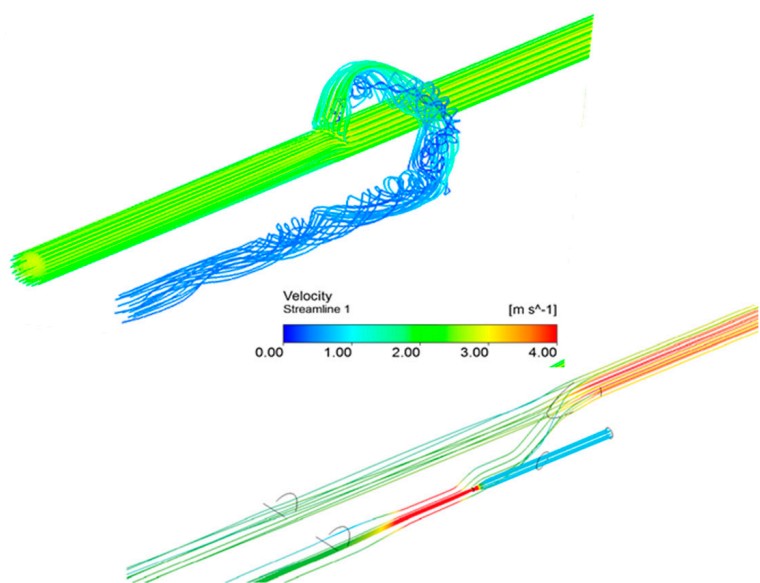

**Figure 24.** Streamline loop with and without pump.

To look at the oil volume fraction, Figure 25 is presented. Even though the eccentricity of the suction chamber cannot be appreciated in this view, the idea was to show the gravity effect, which together with low velocities promotes phases separation and fouling. For the single loop, the phase separation and the effective diameter reduction are clearly shown. That is the fouling that will jeopardize the operational conditions of the gathering system. As expected, adding the jet pump, a homogeneous two-phase flow is obtained. This flow pattern will facilitate the transport of crude oil scaping the fouling formation. Making zoom in oil fraction is shown how the phase separation has been reduced.

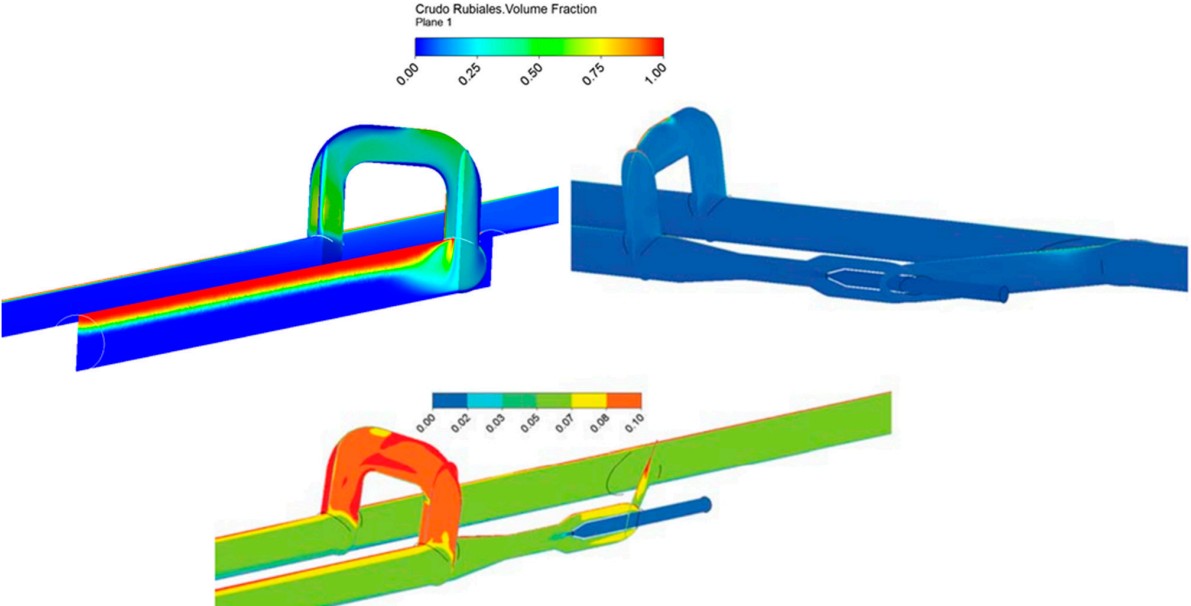

**Figure 25.** Phase distribution in the loop with/without eccentric jet pump.

## 6. Conclusions

The implementation of jet pumps for the transport of heavy crude oil with a high-water cut is a valuable and low-cost effective solution. This type of pump guarantees a homogenous flow pattern avoiding the fouling phenomena and its harmful consequences.

Design and Optimization

1. An integral, fast and efficient methodology for the design and optimization of jet pumps has been presented.
2. This methodology is based on a multiparameter optimization method, and the results obtained by CFD simulations.
3. Before performing experimental evaluations, this methodology allows a time-practical performance evaluation of more than 400 jet pumps models.
4. It was obtained a new optimized pump, with better efficiency than the original one. That could be observed in flow rate, and pressure relations reported.

   Plugin

5. Performance analysis of a trunkline oil gathering system was carried employing CFD.
6. After studying loop behavior without a jet pump, three configurations installing a jet pump were analyzed, the standard connection at 90° and two connections at 30°.
7. Poor performance was obtained for standard jet pump connection, barely reaching 10% of the extra flow.
8. Results show an improvement in total handled flowrate over 30% using either option one o option two.
9. Internal velocity fields presented reinforce these results and delight in assuming uniform flow through the pump could be wrong.
10. The flow rates ratio between the motive fluid line and the exit of the parallel line seems to be a constant value near 30%.

    Eccentric

11. A practical case of study of fluid transportation has been presented.
12. The addition of an eccentric suction chamber improves the performance of the design and helps to uniformize the flow in the throat and diffuser.
13. The eccentric design achieves up to 17% more total handled flowrate at a low flow rate from the high-pressure inlet ($Q_4 \approx 3\%$) with the eccentric shell. This result shows an improvement between 3% and 7% compared to the concentric one.
14. The eccentric design obtains a flow distribution of over 40% for all operation ranges, which is better than the obtained with the concentric design.
15. The addition of the eccentric design improves the performance, uniformizes the flow rate in the throat, and helps to make homogeneous the mixture of heavy-oil/water flow.

## 7. Recommendations

1. It is recommended to validate the performance of the jet pump for proposed options experimentally.
2. It is recommended to study another shape of the throat design to maximize the contact area of the primary and secondary flow. This could help to make a more efficient mixture and reduce the size of the equipment.

**Author Contributions:** Conceptualization: R.N. and M.A.; methodology, M.A. and J.T.; software, J.T.; validation, J.T., R.N. and M.A.; formal analysis, J.T., R.N. and M.A.; resources, R.N. and M.A.; writing—original draft preparation, M.A.; writing—review and editing, M.A.; supervision, R.N. All authors have read and agreed to the published version of the manuscript.

**Funding:** This research received no external funding.

**Data Availability Statement:** Data sharing is not applicable to this article.

**Acknowledgments:** The authors gratefully acknowledge Frontera Energy for having supported this research.

**Conflicts of Interest:** The authors declare no conflict of interest.

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
