# Peer review of "New Design and Optimization of a Jet Pump to Boost Heavy Oil Production"

_computation, doi:10.3390/computation10010011_

Round 1
Reviewer 1 Report
The paper presents an approach to designing and optimizing the use of jet pumps in pipelines for heavy oil using loops. The paper is well structured and easy to follow with sufficient language quality. There are some typos here and there and it should be carefully proofread.
Some comments for authors to address:
- not sure if this was affected by editorial preparation of the paper but the figure captions within the text have been lost (so, not necessarily an issue for authors)
- Figures 4 and 6 have been published previously so it should be addressed by proper referencing
- in section 3.2. there is no real need for showing generally known information about how the optimization process work, but to simply stick to the points relevant to the problem (simply define the method being used and the parameters being optimized)
- section 4.1. lines 204/205: revise the use of fine.. fine mesh, fine grid and medium grid is mentioned which should be consistent
- was there any comparison of the initial CFD calculation done to validate used setup (before optimization part)?
- line 269: reference error (probably figure caption)
- in the section 5 where the recirculation is mentioned, it could be presented better by using streamlines and vectors to show the direction of the fluid
- it should be described that is the difference between options 1 and 2 for concentric (note the typo for option 2 in the chart) and eccentric pumps (also a typo in the chart)
- looking at charts show in Figure 20 alone, there is no clear advantage of using eccentric setup. How was the position of eccentricity selected? Can this parameter also be optimized?
- In Figure 25, the fact that it is using an eccentric setup is not clearly visible (looks pretty concentric) so perhaps another view from the top might be beneficial to remove the doubt. Also the color bar legend should clearly state which volume fraction is being displayed
Author Response
"Please see the attachment."

Reviewer 2 Report
In this study, computational fluid dynamics was adopted to optimize the configuration of jet pumps in pipe loops for heavy oil transport lines.
The paper is well written, the method used seems to be proper, the results obtained are reliable, the conclusions sound reasonable.
It is suggested to accept this paper with minor revision,
- Check the context carefully, and correct the errors that occurred in the pictures and references cited in the text.
- for the optimization process shown in fig 8, are the optimizations of different parameters done simultaneously or separately?
- Is there a flow mechanism corresponding to optimization, for example, what improvement has been made in fluid flow after optimization?
Author Response
"Please see the attachment."
